

# Students' performance in interactive environments: an intelligent model

Doaa Mohamed Elbourhamy[1], Ali Hassan Najmi[2] and Abdellah Ibrahim Mohammed Elfeky[1,3]

[1] Kafrelsheikh University, Kafrelsheikh, Egypt
[2] King Abdulaziz University, Jeddah, Saudi Arabia
[3] Najran University, Najran, Saudi Arabia

## ABSTRACT

Modern approaches in education technology, which make use of advanced resources such as electronic books, infographics, and mobile applications, are progressing to improve education quality and learning levels, especially during the spread of the coronavirus, which resulted in the closure of schools, universities, and all educational facilities. To adapt to new developments, students' performance must be tracked in order to closely monitor all unfavorable barriers that may affect their academic progress. Educational data mining (EDM) is one of the most popular methods for predicting a student's performance. It helps monitoring and improving students' results. Therefore, in the current study, a model has been developed so that students can be informed about the results of the computer networks course in the middle of the second semester and 11 machine algorithms (out of five classes). A questionnaire was used to determine the effectiveness of using infographics for teaching a computer networks course, as the results proved the effectiveness of infographics as a technique for teaching computer networks. The Moodle (Modular Object-Oriented Dynamic Learning Environment) educational platform was used to present the course because of its distinctive characteristics that allow interaction between the student and the teacher, especially during the COVID-19 pandemic. In addition, the different methods of classification in data mining were used to determine the best practices used to predict students' performance using the weka program, where the results proved the effectiveness of the true positive direction of functions, multilayer perceptron, random forest trees, random tree and supplied test set, f-measure algorithms are the best ways to categories.

## INTRODUCTION

In recent times, Egypt has gone through exceptional circumstances due to the COVID-19 pandemic that affected all fields, especially the educational field. Being the corner stone of the learning process of any country, it is ultimately essential to have new generations capable of production and advancement. As part of the precautionary measures, all educational institutions need to limit the spread of the coronavirus and turn to self-learning and virtual e-learning (*Alhalafawy , 2021*; *Alhalafawy & Zaki, 2022*; *Sahrir, Zainuddin &*

Corresponding author
Abdellah Ibrahim
Mohammed Elfeky,
abdalah.elfeqi@spe.kfs.edu.eg

*Nasir, 2016*). Hence, e-learning management systems use infographics in web-based e-learning environments in an attempt to minimize infection (*Afify, 2018*; *Barhoumi, 2015*; *Nedeva & Pehlivanova, 2021*; *Tay & Allen, 2011*; *Wang, Tang & Zhou, 2012*). One of the educational platforms that has spread recently *via* the Internet is a Moodle (Modular Object- Oriented Dynamic Learning Environment) platform, which allows the greatest amount of interactivity and virtualization within the educational environment due to the various means of communication, evaluation and activities. It stimulates and attracts students to stay for a long period of time in front of an electronic device in order to study an educational material such as animated infographics that has the benefits of speed attraction and presentation capabilities needed by students and cannot be fulfilled in real environments. Infographics have a number of benefits, including the ability to simplify scientific facts and present them in a visual data format, the ability to reduce explanatory documents, diagrams, and videos to expressive symbols, pictures, and clear denotations, and the ability to be generated more quickly (*Arif et al., 2021*; *Martix & Hodson, 2014*; *Mohammdi & Elbourhamy, 2021*; *Smiciklas, 2012*; *Vanichvasin, 2013*).

Infographics can be designed in more than one type; and perhaps the most prominent ones are static and animated types. Static infographics are graphics that are intended for print or digital use in websites, or to be displayed on a screen as digital displays, but do not have any motion, animated components, or motion assets. Animated infographics are graphics created to be viewed on graphic animated video screens on video websites like YouTube, TV commercials, or animated presentations on smartphones. The elements and data in animated form are constantly moving and are distinguished by a great deal of ingenuity in the selection of expressive movements which aid in the creation of an engaging and enjoyable animation. Furthermore, this type necessitates a complete scenario in order to create the final form using the required software programs (*Alshammary & Alhalafawy, 2022*; *Alzahrani & Alhalafawy, 2022*; *Hassan, 2016*; *Lankow, Ritchie & Crooks, 2012*). In a variety of fields, many types of data are generated and obtained. Data must be collected, organized, and analyzed in order to extract valuable information. Real-world domains and sectors must analyze massive quantities of data to gain useful information. Data mining techniques are used to construct a model that analyzes the dataset and finds useful patterns. Data mining (DM) is a collection of analytical techniques for analyzing data and extracting useful information.

Educational data mining (EDM) is a new data mining discipline. The main goals of EDM are to collect data from learners and their learning environments, and to propose new methods for identifying useful trends in this data that tends to be useful in understanding learners and learning environments and factors that link the two together (*Meghji et al., 2020*; *Mohammed, 2020*). Educational institutions are still eager to gather information about their pupils. The extensive processing of this data will reveal areas in which the organizations need to develop. With the emergence of data analysis and the increased acceptance of online learning environments, there has been a surge of interest in data collection and processing. Data collection and review from students and their surroundings appear to be beneficial in assisting and gaining insight into students' learning practices (*Hussain et al., 2018*; *Jauhari & Supianto, 2019*; *Kaur, Singh & Josan, 2015*; *Khan et al., 2019*; *Shahiri & Husain, 2015*).

Because of the use of interactive virtual environments, there is a need to track students' performance and predict the speed of their progress in order to improve their performance. Machine learning algorithms can be used to predict students' performance. In the current study, animated infographics are used to demonstrate the technique of computer networks learning by the movement of data in wires and the different signals guided to network devices. This study has merged the use of animated infographics in teaching computer networks for the fourth-year computer students' teachers. Data mining was used to forecast the performance of those students while teaching them through a series of videos built with infographic technology to demonstrate how computer network functioned from the inside due to the inability of seeing it in the conventional environment. Indeed, on the Moodle platform, students' success was projected by showing a questionnaire that asked students a series of questions about infographics and the degree to which students benefited from presenting science content using infographics in general, as well as assessing the efficacy of teaching with infographics on students. The efficacy of other relevant social and academic data on students using the various classification methods within the datamining and the best strategies was ensured in order to predict the student's execution through the Weka program at the beginning of the second semester of the academic year (2019/2020).

## LITERATURE REVIEW

### Animated infographics (AIS) and cloud computing services (CCS) in learning

The use of AIS and CCS has great advantages in the process of learning at a time when the ratio of learners to teachers is increasing at the national and global levels, simply as it aids students to understand a particular program or topic and increases data and information retention over time (*Siddiqui et al., 2019*). In addition, high costs can be avoided since application updates are available immediately like e-learning platforms that are hosted in the cloud and are secure and improve employee performance and efficiency (*Elfeky & Elbyaly, 2019*; *Paul & Das, 2016*). Meanwhile, they facilitate seamless collaboration among dispersed staff, paper compatibility that has been improved. Also, there is no need for internal IT support in order to encourage employees to stay with the company since the e-learning systems that are hosted in the cloud are dependable (*Alavi & Mohan, 2013*; *Subramanian et al., 2014*). The use of AIS and CCS have great advantages from the perspective of students who are enrolled in an online course, can submit homework or tasks, projects, provide reviews, and complete exams (*Elfeky & Elbyaly, 2017*). Moodle provides useful and motivating tools for online and in-class teaching, independent and individual work to be more effective (*Masadeh & Elfeky, 2016*). As per the present study, the features of the educational platforms (Moodle) as well as the features of the (AIS) have been used. Figure 1 illustrates how cloud-based e-learning works in the current study.

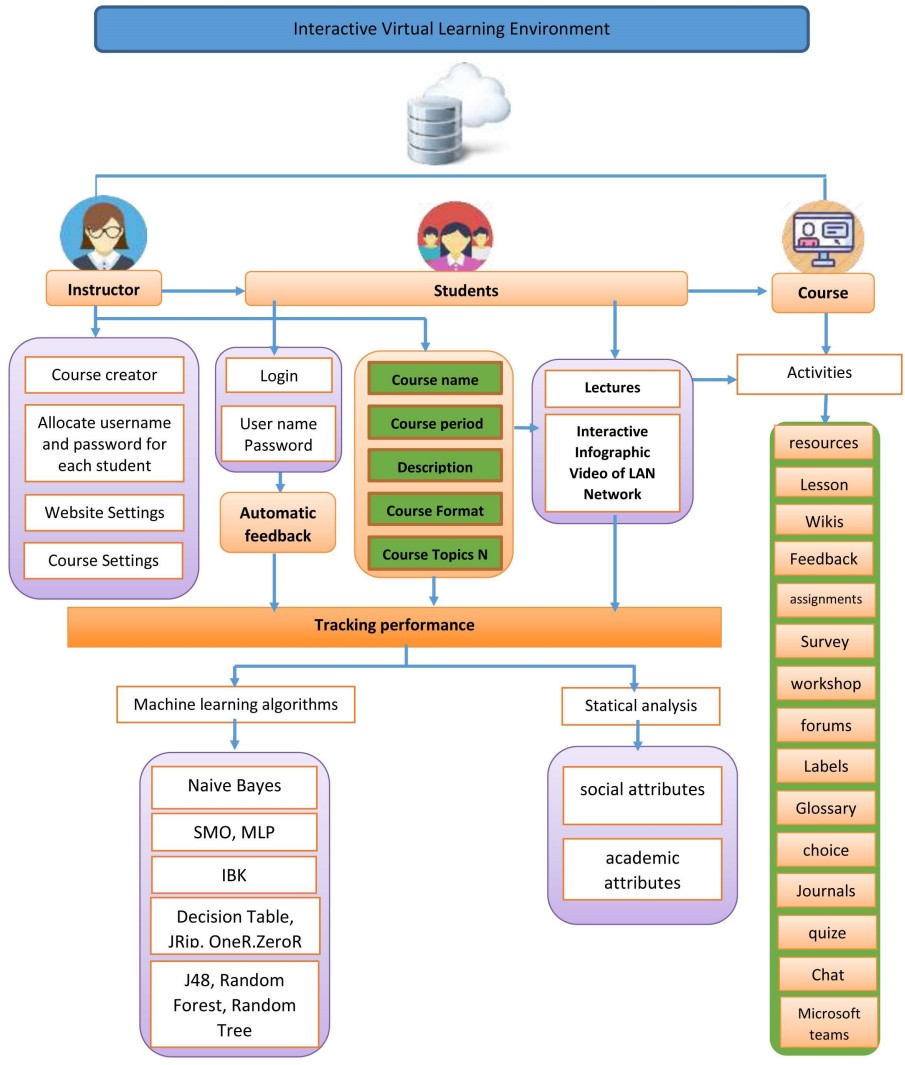

**Figure 1** Block diagram of track student performance in interactive environments.

## Machine learning and tracking students' performance

There is a wide variety of EDM-related work available, with many intriguing methods and resources aimed at achieving the goals of exploring information, making decisions, and making recommendations. These will be reviewed below to show how far they helped us with this current study. It can be seen in a study on the application of big data in the educational field, *Elfeky & Elbyaly (2021)* that big data techniques, such as performance analytics, can be used in a variety of ways to support learning analytics. Data visualization, intelligent input, course recommendation, prediction, attrition risk identification student ability estimation, activity detection, and student grouping and coordination are only a few of the things that can be done. The utility of predictive analysis, which is oriented to the prediction of student actions, ability, and success, is highlighted (*Alharbi, Ahmed & Elfeky, 2022*; *Kaur, Singh & Josan, 2015*). The learning and educational analytics were investigated

in a study conducted at Northern Taiwan University. The aim of using big data methods was to make an early prediction of the final academic result and the students' ability to pass a calculation course in which the principal component analysis was used and the students' final academic output was predicted by using regression analysis (*Elfeky, Alharbi & Ahmed, 2022*; *Lu et al., 2018*). Table 1 shows a comparison between some previous studies that focus on algorithms and their impact on student performance.

### Summary of the previous studies

A prior review of the previous research revealed:

- The efficacy of employing algorithms to anticipate and enhance students' performance.
- The Weka program's broad usage in forecasting its simplicity of use.
- The efficacy of Algorithms Decision Tree (J48), adaBoost SAMME, and adaBoost for prediction in a variety of research, particularly those pertaining to the educational process.

Therefore, the role of both AIS and CCS in different educational fields has emphasized the importance of using machine-learning algorithms in the follow up of students' performance in the future educational stages and predicting their performance, and the dearth of research that integrates machine learning algorithms and infographic within a free Moodle educational platform. After reviewing previous analysis, it is apparent that infographic technology should be integrated with data mining techniques to forecast students' success. Consequently, the current study is focused on monitoring students' performance using machine learning techniques and forecasting improvement in their performance using animated infographics technology in virtual environments. From previous studies, the current research used one or more algorithms from each category available in the Weka program to prove the effectiveness of the best algorithms for predicting student performance.

## PROPOSED FRAMEWORK

In the present study, cloud computing Software - as-a-Service (SaaS) was used. These application software facilities are provided to the users on demand. Users need not worry about its installation, setup, and running of the application because service provider will take care of it such as Moodle that used in this study. Through this type, a new Virtual Cloud Learning Environment (VCLE) system was designed to improve students' performance in the computer networks course using the model platform most used in the educational field. A set of videos were produced with infographic technology explaining the computer networks course, as the network course required a lot of the student's effort to understand and imagine what was happening inside the network and how the data was passed in the form of electrical signals in the network wire. Computer as the main content that was presented through cloud computing technology, due to the emergence of the coronavirus, so the Moodle platform was used so that students could obtain scientific material online through the learning platform with the availability of various interaction methods.

**Table 1 Comparison between related workies.**

| System | Method/Algorithm | Corpus | Results |
|---|---|---|---|
| Decision tree learning used for classification of student archetypes in online courses | They present six decision trees for classifying the finalization and participation rates of online courses based on the students individual traits. | A total of 632 students from Romania regarding the advantages and disadvantages of MOOCs | Students enrolled in paid courses will have a very high rate of dedication and finalization; post-graduate students are four times more dedicated to eLearning than undergraduates; similarly, male post-graduates are two times more likely to participate than their female counterparts. Also, the desire for free courses is omnipresent, as well as certification, except those students only interested in self-development and diversity, which do not usually pursue acertification. |
| Tracking Student Performance in Introductory Programming by Means of Machine Learning | They applied 11 Machine Learning algorithms (from 5 categories) | Dataset contains complete record of 50 students is stored in the format required by WEKA. | Decision Tree (J48) is giving higher accuracy in terms of correctly identified instances, F-Measure rate and true positive detections. |
| Predicting Student Performance in Higher Education Institutions Using Decision Tree Analysis | Three built classifiers (J48, Random Tree and REPTree) were used in this model with the questionnaires filled in by students | The survey consists of 60 questions that cover the fields, such as health, social activity, relationships, and academic performance, most related to and affect the performance of students. A total of 161 questionnaires were collected. | J48 Decision Tree is able to reach an effectiveness of 88% when the students have taken their first exams (15% of overall grade). |
| Building student's performance decision tree classifier using boosting algorithm | They propose three boosting algorithms (C5.0, adaBoost.M1, and adaBoost.SAMME) to build the classifier for predicting student's performance | This research used 1UCI student performance datasets | The result of the first scenario showed that adaBoost.SAMME and adaBoost. The second scenario was used to evaluate boosting algorithms under the different number of training data. The third scenario, we build models from one subject dataset and test using another subject dataset. The third scenario results indicate that it can build prediction model using one subject to predict another subject. |
| An Improved Prediction System of Students' Performance Using Classification model and Feature Selection Algorithm | Linear classifier (Logistics regression, Naïve Bayes classifier, Fisher's linear discriminant), Support Vector Machine, Quadratic classifier, Kernel estimation (K-nearest neighbor), Decision tree (Random Forest), and Neural Networks (Learning vector quantization) machine learning algorithm. | Dataset contains 33 attributes and 649 instances. | In the feature selection algorithm, WrapperSubsetEval method with Random Forest classification is better than other feature selection method and other classification algorithm to predict students' performance |

**Table 1** (*continued*)

| System | Method/Algorithm | Corpus | Results |
|---|---|---|---|
| Students' Performance Analyses Using Machine Learning Algorithms in WEKA | BayesNet (BN), Multilayer Perceptron (MLP), Sequential minimal optimization (SMO) and Decision tree (J48). | A survey was compiled, which included 32 questions. The survey was provided to 115 students studying in 2 engineering programs at the Trakia University - Stara Zagora. | The obtained results show that the MLP algorithm is the best for the used data. The obtained accuracy is sufficient to create an effective forecast model. 12 attributes have been identified that have the greatest impact on student performance. |

After designing the infographic and uploading it to the Moodle educational platform, students joined the networking course in the second semester of the academic year 2019/2020. After completing almost half of the course, the next step was to use a questionnaire to measure the effectiveness of the study on the educational platforms by predicting the performance of students after presenting almost half of the educational content to improve the students' performance in the coming stage when completing of the course. In the current study, cloud-based e-learning worked as shown in Fig. 1. Figure 1 shows how cloud-based e-learning performed in this project. Figure 1 shows the proposed system, which allows the system administrator to upload learning resources such lecture slides, AIS, assignments, quizzes, audio, and video. Depending on the nature of the subject, each lecturer can assign homework as well as the learners' requirements. The activities and materials are only available to students who have enrolled for the module. They can turn in assignments, send and receive messages, attend seminars, and make changes to the wiki. Professors can use the technology to design and run a virtual class while simultaneously providing feedback to students. An IT professional or a system administrator teaches the majority of Moodle courses.

Therefore, the suggested system is intended to benefit educational institutions by combining beneficial services that encourage users to complete work correctly and efficiently. As a result, the study includes a number of AIS videos to teach pupils more in-depth computer expertise. Networks, the content shows students how the networks work, and the data passes inside the wires down from the sending device to the receiving device that they cannot imagine or see in the traditional learning method. In order to predict students' performance from AIS technology, the researcher utilized knowledge discovery and data mining methodology in predicting student's academic performance. Therefore, the proposed system is divided into two subsystems: the VCLE system and Track Student Performance model.

## Participants

The sample comprised of (300) students from Kafrelsheikh University's Department of Educational Technology–Computer Teacher Preparation. The students were assigned to a computer network class course. The main topics that were raised in the proposed system are shown in Fig. 2 that displays a map of computer networks course contents. Their standard deviation of their age was 2.14 and the average age was 17.9 years. All participants were informed of the research, and signed forms of consent before the course proceeded (*Elbyaly*

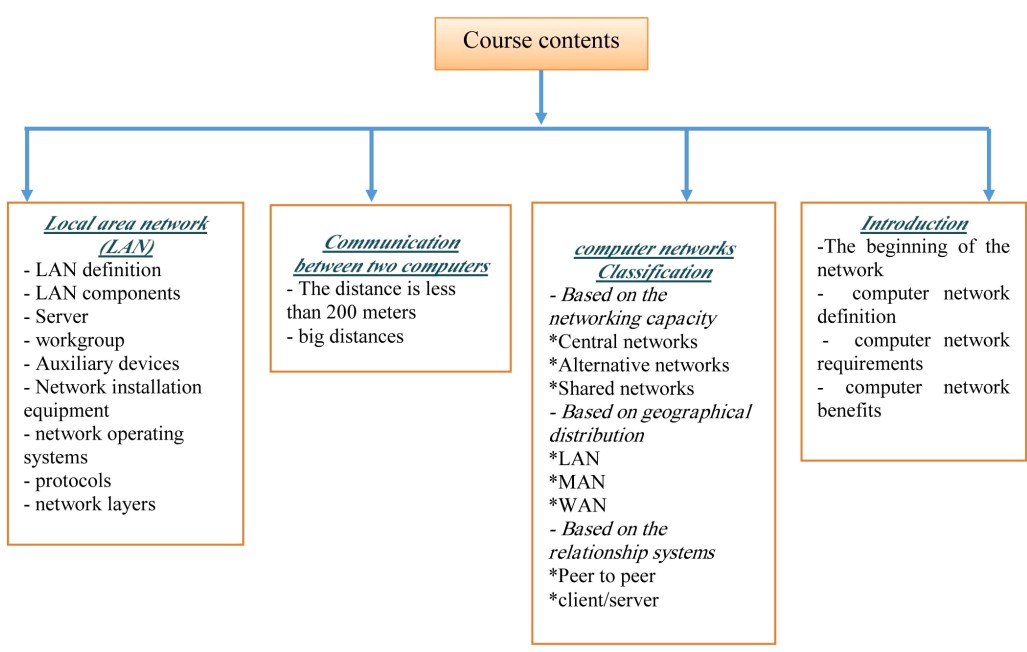

**Figure 2** Map of computer networks course contents.

*& Elfeky, 2022*; *Elfeky, 2017*). We gave them the chance to withdraw/not participate without penalty.

## VCLE system architecture

The following components are included in the proposed system:

– Users: The users of the VCLE system are students and lecturers who are in charge of producing, organizing, and regulating learning materials, exercises, and content.
– VCLE: The VCLE is a new Moodle system that offers flexibility in terms of where services are located and how students can use them. It also allows users to monitor activity as well as choose and arrange data.

### VCLE system structure

(A) Creating a website for the computers network course and adjust all the related settings such as the course full and short name, course start and course end, description, appearance, and course format.
(B) Adding the course's objectives and sub-objectives to the system for each lecture. To access the system and browse all of the components, each student was given a login and password. Lectures, AIS, single activities, quizzes, and exams, drag and drop into text, drag and drop into an image, yes or false, and other forms of course subjects were divided. These provided students with automatic feedback.
(C) Each lecture included the following:

– Materials for the lecture (written, PowerPoint slides, AIS videos, pictures, tables, and charts)

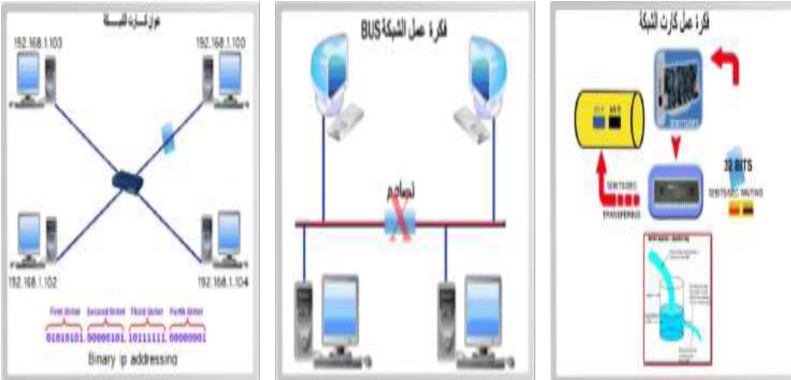

**Figure 3  Screenshot of the AIS videos.**

– AIS that brought the subject to life and helped students to imagine what would happen in a real-world scenario Fig. 3 shows a selection of AIS footage screenshots. Figure 3 depicts the VCLE system.

(D) To fill in the blank phrase, several questions, such as multiple-choice, are employed, as well as drag and drop int false, and more options are available. After each lecture, the students were instructed with automated feedback.

(E) Simultaneous chat rooms. Students might speak with their professor about concerns concerning the lectures or the system. Assignments, workshops, and a wiki were also accessible in the system.

(F) Quizzes. After each lesson, students could practice on many quizzes, and the technology provided appropriate automated feedback.

(G) Information about the student's most recent access to the course, current status, and grades. The system could report for each student individually or all students.

(H) Exams: After completing the learning procedure required for each objective, students could take the test and receive their final result. All grades were saved in the system, and the lecturer could save and check all of the students' responses to keep track their advancement.

Figure 3 shows snapshots of a selection of videos produced using infographic technology, which illustrates part of the networking course used in the current research (network card address- network bus idea- network card idea)

As shown in Fig. 4, the VCLE can help in tracking students' performance through its systematic steps.

Figure 4 depicts a snapshot of the Moodle application after integrating the networking course content utilized in the current study.

The intelligent model for tacking students' performance includes four stages as shown in Fig. 5.

Figure 5 shows the stages of the student performance tracking system, which will be explained in the following points.
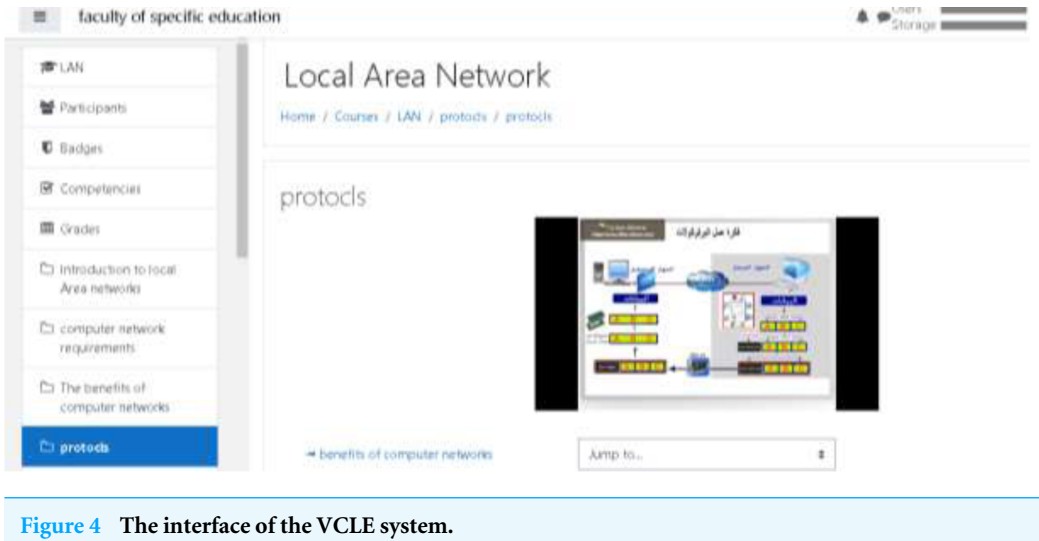

**Figure 4** The interface of the VCLE system.

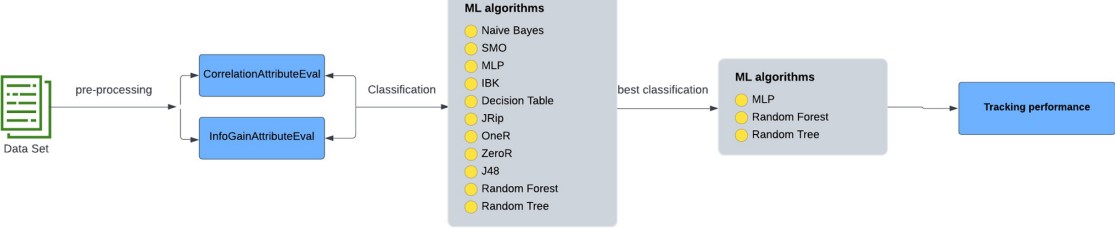

**Figure 5** Diagram of intelligent model for tracking students' performance.

### Data gathering

This dataset of student's scores at the Computer Networks course was obtained from fourth-year students in the Computer Teacher Department at Kafrelshiekh University in the second semester of the academic year (2019/2020). An online questionnaire, generated using Google forms, was used to collect the necessary data (https://forms.gle/iryvrdCHqHVmjabt7). The questionnaire (*Alharbi, Elfeky & Ahmed, 2022*; *Su, Liu & Wang, 2018*) was completed by 300 students and provided a sufficient dataset size. The data collection form was divided into two sections, the first of which was an infographic-based questionnaire on students' opinions of teaching. The attributes used to categorize students' success (cumulative GPA) were divided into four categories: excellent, very good, good, and fail. The second section included social and academic information about the student, which was divided into social characteristics such as gender and family size, as well as academic characteristics such as behavior, study time (daily), project, and GPA. Table 2 shows the information for this dataset.

Some difficulties with the dataset might impair the accuracy of the ML algorithms; therefore, the dataset was inspected for producing a better model. Because of this, the researchers employed several methods given by Weka to pre-process the data (described

**Table 2  Dataset attributes.**

| Attribute | Description | Possible Value |
|---|---|---|
| | First part | |
| Q1 | This infographic is helpful for me. | |
| Q2 | This infographic is helpful to Spread the Culture of Educational network | |
| Q3 | This infographic is more instructive compared to long texts. | |
| Q4 | This infographic is helpful for Educational activities. | |
| Q5 | This infographic makes it easier for me to understand the Educational courses | |
| Q6 | This infographic saves my reading time | |
| Q7 | This infographic is easy to use. | |
| Q8 | This infographic is friendly to user. | |
| Q9 | This infographic is flexible. | Strongly agree = 5, |
| Q10 | I can use this infographic without written instructions. | agree = 4, |
| Q11 | I can use this infographic successfully every time. | natural = 3, |
| Q12 | I can learn to use this infographic very quickly. | disagree = 2, |
| Q13 | I can easily remember how to use this infographic. | strongly disagree = 1 |
| Q14 | It is easy to learn to use this infographic. | |
| Q15 | I quickly became skillful with this infographic. | |
| Q16 | This infographic, presentation of information within a context facilitates learn network. | |
| Q17 | I am very satisfied with this infographic. | |
| Q18 | I would recommend this infographic to my friend. | |
| Q19 | It is fun to use this infographic. | |
| Q20 | I feel I need to have this infographic. | |
| | Second part | |
| Gender | Student's gender | Male, Female |
| Age | Student's age | 20–26 |
| *F*-size | Family size | Student's family size less or equal or more than 3 |
| Health | The health status of students | The health status of students (from 1 to 5) |
| Internet | Internet presence | Internet presence |
| S-Time (Daily) | Study time (daily) | 1 to 5 h |
| Activity | Activity acceptance | Yes or no |
| project | Project acceptance | Yes or no |
| GPA | Cumulative GPA | >50 □ weak <br> <= 50 to 65 □ fair <br> <= 60 to 75 □ good <br> <= 75 to 85 □ very good <br> <= 85 to more □ excellent |

later in this study). First, the online questionnaire data was exported to an Excel file. The various sorts of data were then examined and, if necessary, updated. Our dataset will be ready for machine learning implementation by the end of this point. The types of data were

then reviewed and, updated if required. A well-known machine learning toolbox is the Weka toolkit (*Alsalman et al, 2019*). It uses Java to build a wide range of cutting-edge machine learning techniques. Weka comprises, among other data mining techniques, algorithms for regression, grouping, clustering, and association laws. It also features user-friendly data pre-processing as well as visualization software Weka has acquired popularity in the data mining and machine learning communities due to the fact that it is an open source, well-developed technology.

### Data mining

Weka is an open-source software which consists of a collection of machine learning algorithms for data mining tasks. Therefore, the largest number of classification algorithms available in the Weka program was used in all groups to extract the best algorithms that predict students' performance. The completed dataset was put into machine learning algorithms, which may then be used to anticipate student academic achievement. The researchers picked 11 machine learning algorithms from five categories: Bayes (naive Bayes), Function (SMO, multilayer perceptron (MLP), Lazy (IBK), Rules (decision table, JRip, OneR, and ZeroR), and Tree (J48, random forest and random tree). The algorithms were all run, tested, and compared.

### Interpretation

Finally, the collected data will be used to create a model for predicting academic success of students. The researchers built the intelligent model with the most accurate algorithm possible.

## Removing high dimensionality

Because our data source has a high degree of dimensionality, we tested Weka's attributes selection methods on it. We used feature selection to reduce the number of overlapping features, prevent over fitting, and potentially increase the feature set's predictive accuracy. We combined Ranker search techniques with the selection of feature subsets based on correlation (CorrelationAttributeEval) and information-gain (InfoGainAttributeEval) (*Khan et al., 2019*).

Table 3 highlights the results of both algorithms. The value 0.20 (Average attribute values in InfoGainAtribute and CorrelationAttri) is kept as cut-off significance value and ignore attributes lesser than this value. Table 3 illustrates that Project and Q16 was shown vital in both the algorithms. CFSSUBsetEval with the GreedyStepwise search tool were used which provided with important attributes such as Project, operation, Health, S-Time, and Q16. From the above table, it can be deduced that Project and Q16 were the best predictors. In addition, CFSSUBsetEval emphasized the Q4 and Q3 attributes. Finally, as a collection of attributes, the researcher chose GPA, Project, Operation, Q16, Fsize,Q2, Health, Q4 and Q3 to be used in prediction model.

## Ethical statement

The ethical committee in our Kafrelsheikh University have reviewed the study protocol and ethically approved the study under the reference No.:334-38-44972-SD.

**Table 3  The attribute selection.**

| Attribute | InfoGainAtribute | CorrelationAttri | CFSSUBsetEval |
|-----------|------------------|------------------|---------------|
| Project | 0.452 | 0.199 | (?) |
| Activity | 0.140 | 0.092 | (?) |
| Q16 | 0.094 | 0.302 | (?) |
| F-Size | 0.092 | 0.288 | (?) |
| Q2 | 0.111 | 0.266 | (?) |
| Health | 0.122 | 0.265 | (?) |
| Q4 | 0.080 | 0.219 | (?) |
| Q3 | 0.037 | 0.203 | (?) |
| Q6 | 0.069 | 0.192 | |
| Q19 | 0.053 | 0.189 | |
| S-Time | 0.097 | 0.168 | |
| Internet | 0.096 | 0.168 | |
| Q1 | 0.025 | 0.164 | |
| Q8 | 0.025 | 0.159 | |
| Q18 | 0.047 | 0.150 | |

## EXPERIMENTATION & INTERPRETATION

The data were analyzed in two different ways to ensure the correctness of the results. The first method is using the performance measurement validity through the performance measurement equation in Weka program for comparing different classification methods and extracting the best methods for analyzing the data to predict the performance of students after studying using infographic technology. The second method was using the statistical program SPSS to analyze the results of the questionnaire and extract the frequencies, percentages, mean and standard deviation.

### Performance measurement

The researchers compared the effectiveness of applied classifiers through their $F$-Measure value because it provides a single score that balances both the concerns of precision and recall in one number. $F$-Measure (1) is the harmonic mean between precision (2) and Recall (3), as described below (*Elbyaly & Elfeky, 2022*):

$$f - measure = 2 \left( \frac{precision * recall}{precision + recall} \right) \tag{1}$$

Where:

$$precision = \frac{TP}{FP + TP} \tag{2}$$

$$recall = \frac{TP}{FN + TP} \tag{3}$$

True Positive (TP): Positive instances, and classified as positive.
False Positive (FP): Negative instances, but classified positive.
False Negative (FN): Positive instances, but classified negative.

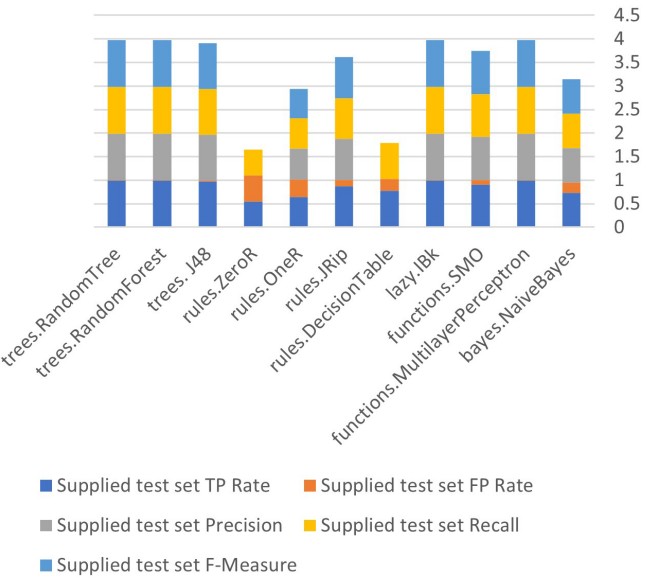

**Figure 6** Algorithm performance of attributes (supplied test set method).

## Algorithm performance before removing attributes

The Weka tool that provides built-in algorithms that sustain in the implementation of different classifiers and get results easily. Eleven algorithms will be used in this stage Bayes (naive Bayes), Function (SMO, multilayer perceptron (MLP), Lazy (IBK), Rules (decision table, JRip, OneR, and ZeroR), and Tree (J48, random forest and random tree) of the three methods used in the current study (supplied test collection, cross validation (10 folds), and percentage split (66 percent) before (less correlated questions).

Figures 6 to 8 show details of performance before removing attributes from the data and further shows the five basic algorithms: Bayes, Function, Multilayer Perceptron, Lazy. Rules, and Tree according to their attributes, namely: True Positive (TP), False Positive (FP) rates, Precision, Recall and $F$-Measure.

In Figs. 6–8, each color in the figure represents the percentage of each True Positive (TP), False Positive (FP) rates, Precision, Recall and $F$-Measure.

## Algorithm performance after removing attributes

Figures 9 to 11 show details of performance after removing attributes from the data and further show the five basic algorithms: Bayes, Function, Multilayer Perceptron, Lazy, Rules, and Tree according to their attributes. Namely, True Positive (TP), False Positive (FP) rates, Precision, Recall and $F$-Measure.

In Figs. 9–11, each color in the figure represents the percentage of each True Positive (TP), False Positive (FP) rates, Precision, Recall and $F$-Measure.

## Results & findings

Weka was used to build the classification model. The evaluation was carried out in Weka (using 10-fold cross validation), with a percentage split (66%) and a supplied test collection,

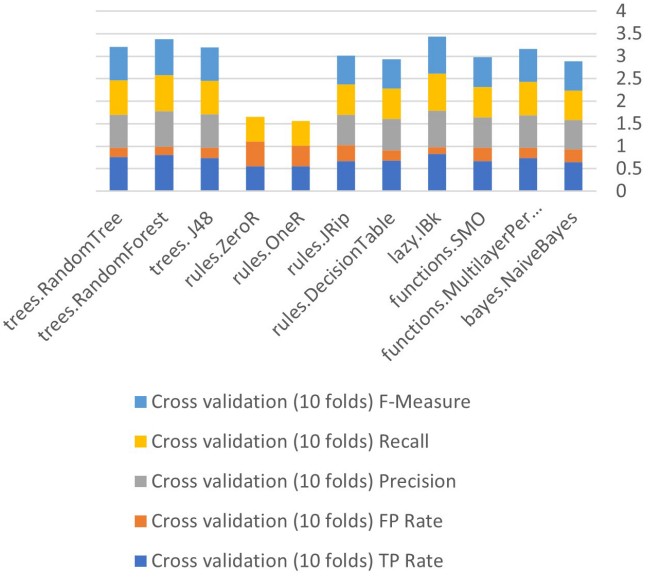

- Cross validation (10 folds) F-Measure
- Cross validation (10 folds) Recall
- Cross validation (10 folds) Precision
- Cross validation (10 folds) FP Rate
- Cross validation (10 folds) TP Rate

**Figure 7** **Algorithm performance of attributes (cross-validation (10 folds)).**

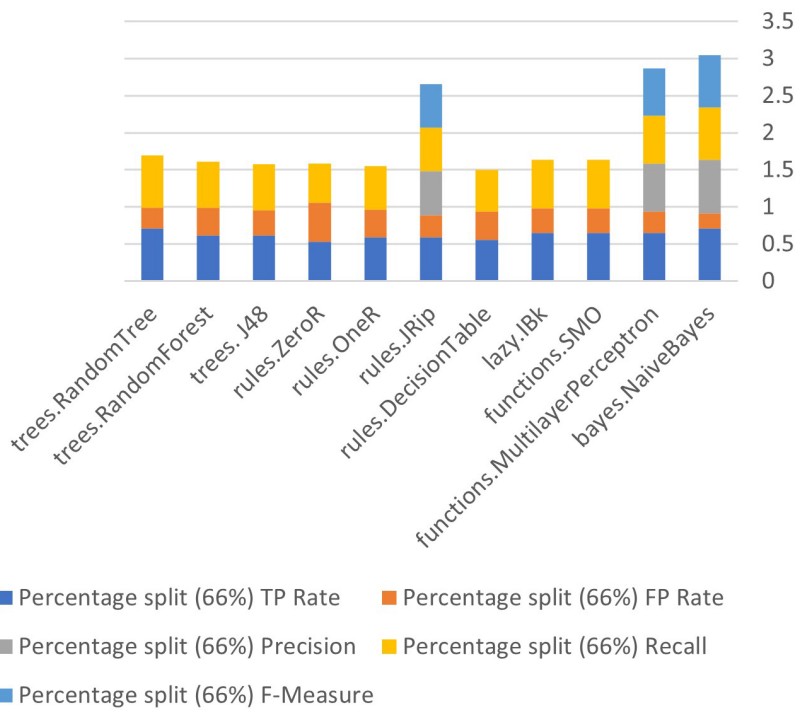

- Percentage split (66%) TP Rate
- Percentage split (66%) FP Rate
- Percentage split (66%) Precision
- Percentage split (66%) Recall
- Percentage split (66%) F-Measure

**Figure 8** **Algorithm performance of attributes (percentage split (66%)).**

both of which are standard practices in predictive data mining applications. After removing the attributes, the researchers compared the output of each classifier. Figure 12 and Table 4 compare the output (percentage of correctly categorized instances) of all classifiers. It has

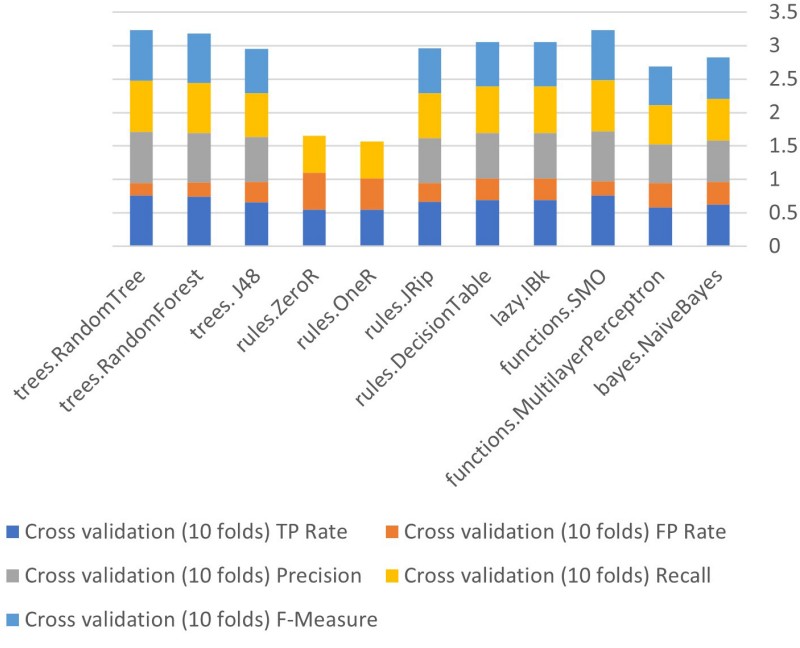

**Figure 9** **Algorithm performance of attributes (percentage split (66%)).**

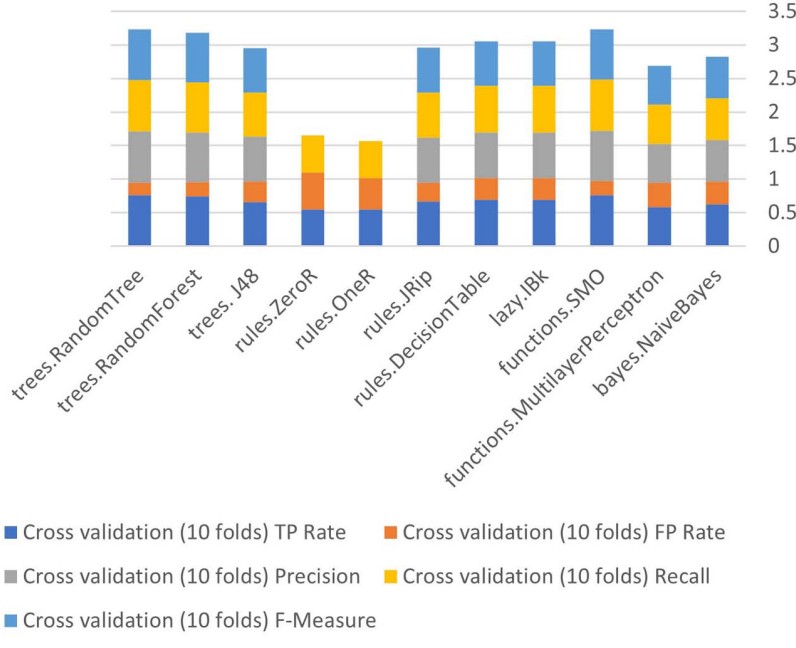

**Figure 10** **Algorithm performance of attributes (cross validation (10 folds)).**

been noted the trees: Trees, RandomForest, RandomTree, as well as its features, Several layers perceptron had a (99 percent) precision in the supplied test set and is also lazy. IBk (83%) in cross validation (10%) or in Bayes' percentage split (66%) Bayes and trees are

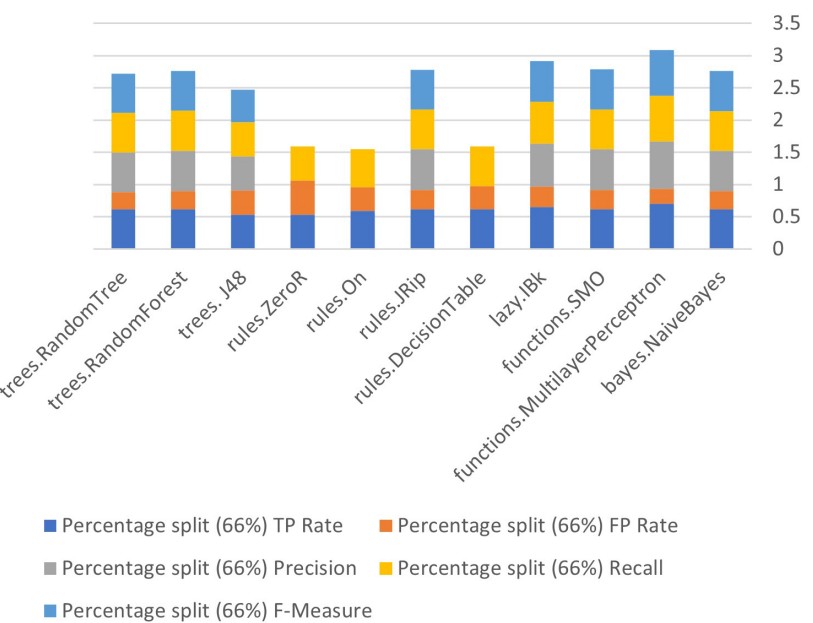

**Figure 11  Algorithm performance of attributes (percentage split (66%)).**

two things that come to mind when I think of Naive. RandomTree reached a 71 percent accuracy rate. Trees, in general. Trees, RandomForest RandomTree, as well as its features, as compared to other types of algorithms by GPA class, MultilayerPerceptron achieved a higher accuracy of (99%) in the supplied test set.

After removing the attributes, the researchers compared the output of each classifier. Figure 13 and Table 5 compare the output (percentage of correctly categorized instances) of all classifiers. It has been found that trees are sluggish. Trees, RandomForest RandomTree, as well as its features. MultilayerPerceptron achieved a 93 percent accuracy in the Supplied Test Set and functions. Trees and SMO as well as trees RandomTree (76%) and works in Cross Validation (10 folds), in percentage split (66 percent), MultilayerPerceptron achieved an accuracy of (71 percent). Trees, in general, belong to the lazy family of algorithms. Trees, RandomForest RandomTree, as well as its features, as compared to other types of algorithms by GPA class, MultilayerPerceptron achieved a higher accuracy of (93%) in the supplied test set.

Generally, the supplied test set was proved to better than cross validation (10 folds) and percentage split (66%) in measuring students' performance. In the supplied test set, when comparing the algorithms before and after deleting the variables, the best three algorithms were functions, multilayer perceptron trees, random forest trees and Random Tree.

## Statistical analysis of the data

The SPSS program was used to perform a statistical analysis of the questionnaire data, which consists of two parts. The first part is a questionnaire about infographic and its importance, and its help for students in studying and comprehension, while the second part contains the students' social and academic data.

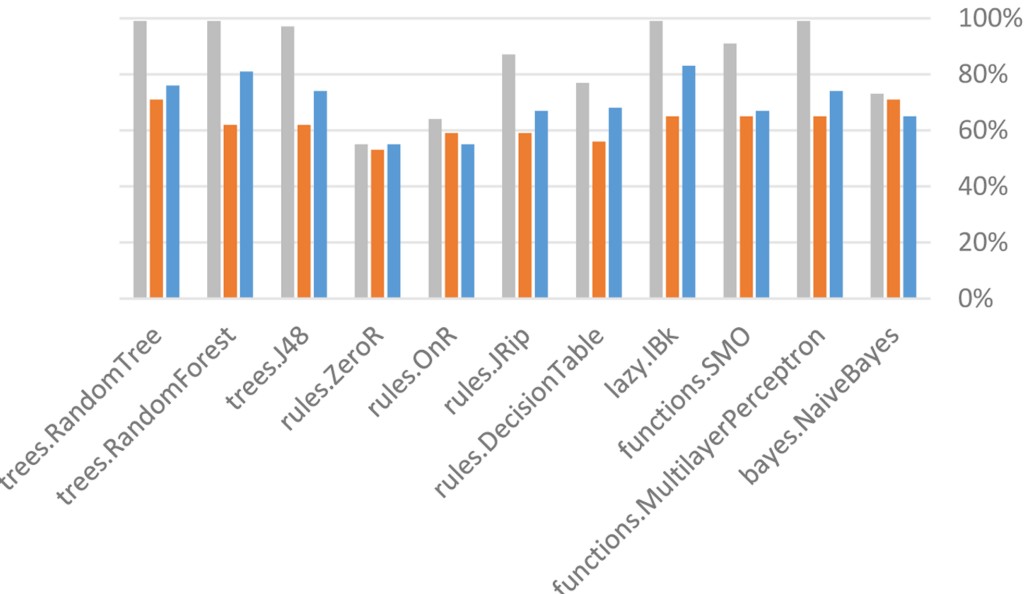

**Figure 12** Accuracy results before the removal of attributes.

**Table 4 Accuracy results before the removal of attributes.**

| Algorithm | Cross validation (10 folds) | Percentage split (66%) | Supplied test set |
|---|---|---|---|
| bayes.NaiveBayes | 65% | 71% | 73% |
| functions.MultilayerPerceptron | 74% | 65% | 99% |
| functions.SMO | 67% | 65% | 91% |
| lazy.IBk | 83% | 65% | 99% |
| rules.DecisionTable | 68% | 56% | 77% |
| rules.JRip | 67% | 59% | 87% |
| rules.OnR | 55% | 59% | 64% |
| rules.ZeroR | 55% | 53% | 55% |
| trees.J48 | 74% | 62% | 97% |
| trees.RandomForest | 81% | 62% | 99% |
| trees.RandomTree | 76% | 71% | 99% |

Figure 14 shows the analysis of the evaluation items for the network course designed with animated infographics, the percentage of frequencies, percentages of standard deviation, and the arithmetic mean of all items. As Fig. 14 shows, Students' responses tend to be "I agree strongly" according to the five-fold Likert evaluation scale.

Figure 15 shows the statistical analysis of the students' social and academic data (the environment surrounding the students). It helps in predicting the performance of the students. The first part of the questionnaire was used for presenting the course items

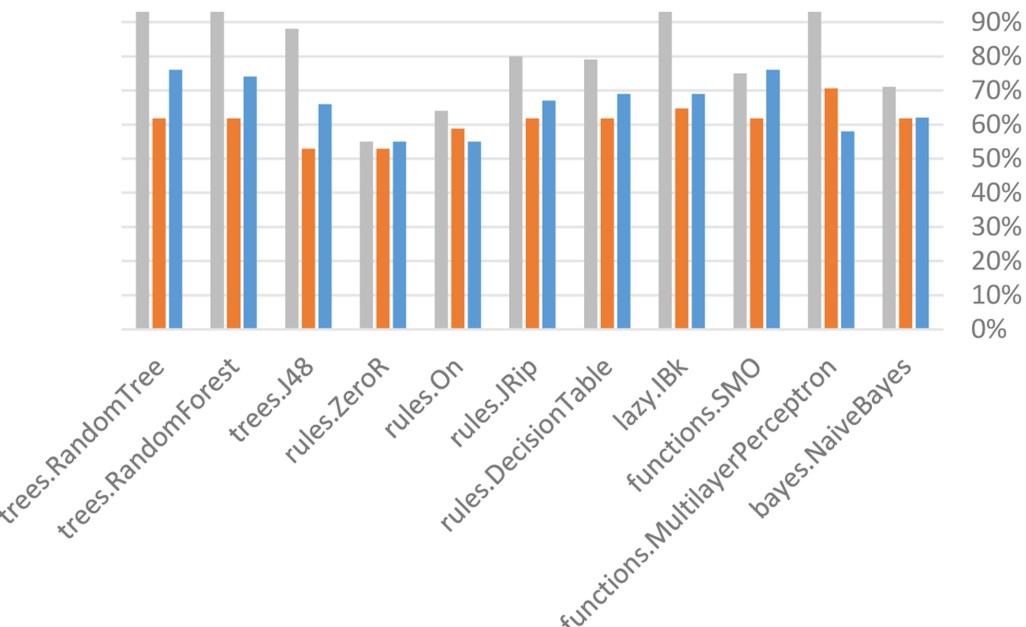

**Figure 13** Accuracy results after the removal of attributes.

**Table 5** Accuracy results after the removal of attributes.

| Algorithm | Cross validation (10 folds) | Percentage split (66%) | Supplied test set |
|---|---|---|---|
| bayes.NaiveBayes | 62% | 62% | 71% |
| functions.MultilayerPerceptron | 58% | 71% | 93% |
| functions.SMO | 76% | 62% | 75% |
| lazy.IBk | 69% | 65% | 93% |
| rules.DecisionTable | 69% | 62% | 79% |
| rules.JRip | 67% | 62% | 80% |
| rules.On | 55% | 59% | 64% |
| rules.ZeroR | 55% | 53% | 55% |
| trees.J48 | 66% | 53% | 88% |
| trees.RandomForest | 74% | 62% | 93% |
| trees.RandomTree | 76% | 62% | 93% |

through the animated infographic technique, where the graph shows the frequencies of each feature of the data as well.

The social and academic data were analyzed according to the students' GPA, especially the grade of excellence, as shown in Fig. 16. It is noticed in Fig. 16 that students who have obtained GPA scores are excellent in personal and study questions phrases:

– As for the FAMILY SIZE item in Fig. 16A, the results revealed the highest value>3. The fewer the family members, the more interest and ability to follow-up education for students increases.

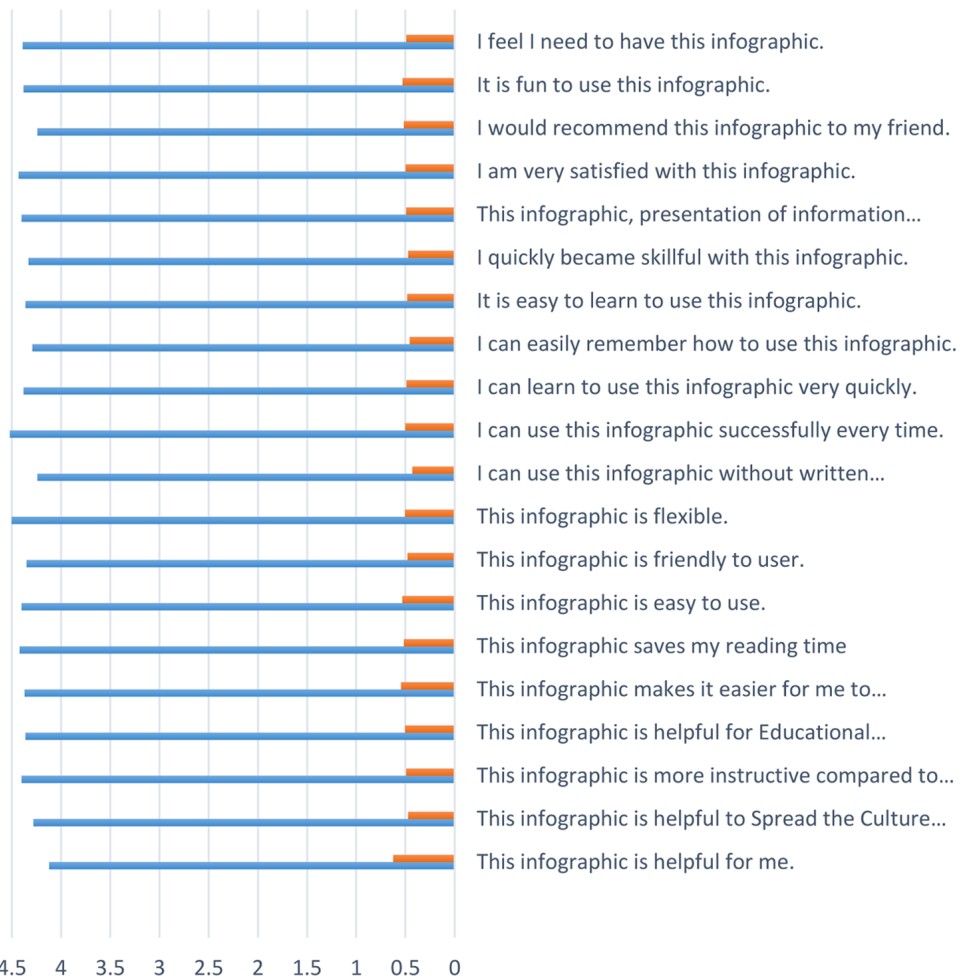

**Figure 14** Analysis of the evaluation items.

– Regarding GENDER in Fig. 16B, the number of female students was more than the number of male students, which indicates the superiority of female students over male students in obtaining a GPA (EXCELLENT).

– Regarding the item of HEALTHY STATUS in Fig. 16C, the number of students who are in good health (4, 5) obtained an EXCELLENT GPA.

– As for the student age item, Fig. 16D shows in an increase in the number of students at the age of 21, followed by 22, 23, because the students to whom the questionnaire was applied were in the fourth year. The other younger students (20), were those who joined the school at a young age, while older students (24, 25, and 26) were students who joined the college after joining one of the other institutes.

– Study time in Fig. 16E results revealed that students who studied 3 to 5 h per day, were the most excelling students.

– Regarding Internet presence in Fig. 16F, the number of students who had internet connection were the ones who got the highest marks, because the scientific content

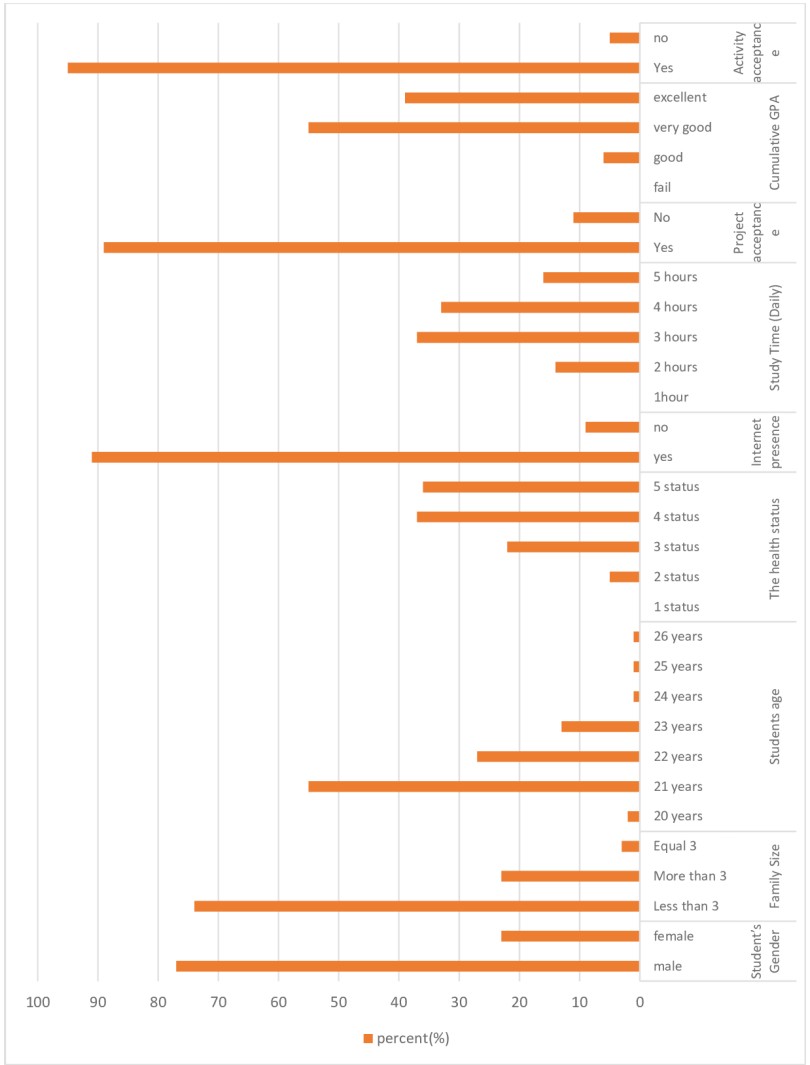

**Figure 15** **Statistical analysis of the students' social and academic data.**

about -computer networks was displayed on the online educational platform Moodle. Therefore, the internet must be available for the students studying.

– Regarding project acceptance in Fig. 16G, the number of students whose project were sent (YES) were the ones who achieved the highest grades.

– Activity acceptance in Fig. 16H the number of students whose activities were sent (YES) were the ones who achieved the highest grades.

## LIMITATIONS AND FUTURE DIRECTIONS

Despite being thorough and rigorous, research does have some limits. First, we solely used data from the Moodle platform for this research. Therefore, in order to maximize the generalizability of the results, we advise future researchers to conduct studies employing data from several platforms. Second, this research project only focuses on one technology

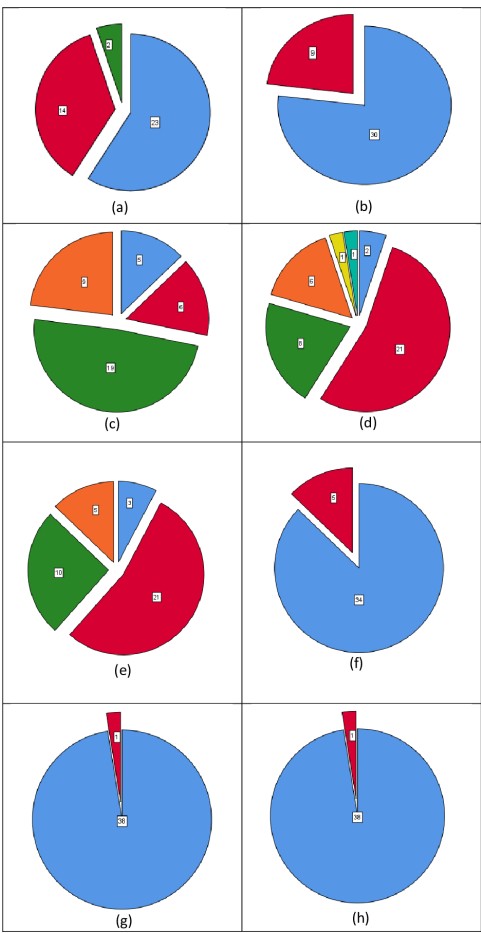

**Figure 16** Analysis of personal and academic data according to the cumulative grade (GPA) of the students, "excellent".

(infographic) inside a single nation; additional research with other technologies may be required elsewhere. Third, this study produced encouraging results about the function of infographic approach as an outside variable on the Moodle platform; future studies may examine the possibilities of enhancing Moodle through its fusion with other outside variables. Finally, since the sample size is perhaps perceived as being limited, additional research with a larger number of diverse fields may be required.

## CONCLUSION

The goal of the present research is developing a prediction model to enable students to notify his potential results, and to evaluate the effectiveness of teaching using Animated Infographics (AIS) by applying a questionnaire in the first half of the second semester. In addition, to find out the students 'views on teaching the networking course using the mobile infographic through the Moodle educational platform. Consequently, classification algorithms were applied to a data source using Weka, and it was deduced that the

algorithms of the supplied test set, functions.MultilayerPerceptron, trees.RandomForest, trees.RandomTree. Algorithms, achieved 93% high accuracy compared to other algorithms. The current study helped prove the effectiveness of teaching using animated infographics based on the Moodle educational platform, taking into account the reasons for the delayed performance of some students after predicting their performance. This helped teachers to improve students'performance in other courses, which leads to reducing the failure rate and improving the requirements for some students in the next semester exams. The statistical analysis of the questionnaire demonstrated the effectiveness of using the infographic of educational platforms (Moodle) in teaching, according to the Likert scale, which improves students' performance when teaching other courses using mobile infographics.

### Funding
The authors received no funding for this work.

### Competing Interests
The authors declare there are no competing interests.

### Author Contributions
- Doaa Mohamed Elbourhamy conceived and designed the experiments, analyzed the data, performed the computation work, prepared figures and/or tables, authored or reviewed drafts of the article, and approved the final draft.
- Ali Hassan Najmi analyzed the data, prepared figures and/or tables, and approved the final draft.
- Abdellah Ibrahim Mohammed Elfeky performed the experiments, analyzed the data, performed the computation work, prepared figures and/or tables, and approved the final draft.

### Ethics
The following information was supplied relating to ethical approvals (i.e., approving body and any reference numbers):

The ethical committee in our University have reviewed the study protocol and approved the study under the reference No.: 334-38-44972-SD.

### Data Availability
The raw data is available in the Supplementary File.

### Supplemental Information
Supplemental information for this article can be found online at http://dx.doi.org/10.7717/peerj-cs.1348#supplemental-information.

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
