# Peer review of "Students’ performance in interactive environments: an intelligent model"

_PeerJ Computer Science, doi:10.7717/peerj-cs.1348_

## Round 0.1 · original submission · Major Revisions

Based on reviewers' comments, the authors are advised to make "major revisions" and resubmit.

Reviewer 1 ·

Basic reporting

1. Moderate usage of English throughout the text (e.g., the message in lines 30-34 is not clear). I recommend the authors to use shorter sentences.
2. The introduction and the body of the text contain thorough references to previous research, which makes it easy to understand connection to related work.
3. I recommend the authors to shorten the abstract – e.g., lines 20-24 can be reduced to a single sentence.
4. I advise the authors to clarify the objective of the research better in Section 1. It is not entirely clear from the introduction/abstract. Is it to explain EDM or to make a prediction of results or to track their current performance?
5. Some of the information is repetitive (such as the names of the ML algorithms in lines 314-332), which are also contained in the tables. Definitions of well-known quantities (e.g., lines 304-311) are provided, which I feel can be omitted while being self-contained.
6. Overall, I think this is a longish paper for the content. For e.g., the comparison between algorithms can be better presented as figures instead of tables.

Experimental design

1. The impact of their research can be followed, although I recommend the authors to clarify this better in the introduction.
2. The authors test against several ML models and then do a feature-selection using the F-measure. However, no clear reasoning or motivation is provided against using accuracy or using a precision-recall curve. I also recommend the authors to shed some light on why they chose these specific ML models.
3. In several cases, the F-score reduces considerably after performing feature-selection; it will be useful to add some discussions on that.
4. Overall, I think that it will be useful to include some ensemble methods beyond Random Forests and to add more explanations rather than just providing numbers in tables.

Validity of the findings

1. There exist many sophisticated techniques for performing feature selection. The authors mention two such in lines 277-278, but no more discussions are added. I highly recommend the authors to do so, as a major part of their work is based on the feature-selection.
2. No discussions are provided on the choice of their ML models, why simpler models like SVM or Logistic Regressions were not tested, why the F-score was chosen to compare before and after feature-selection, etc. I highly recommend the authors to add such discussions.

Additional comments

Nothing apart from the ones mentioned above.

Reviewer 2 ·

Basic reporting

1. In abstract, the scheme of the study is not clear. Instead of using general terms be more
specific.
2. Provide appropriate literature in order to support your work more particularly in terms of
machine learning algorithms don’t describe them in general terms be more specific.
3. Give appropriate abbreviation and explanation before using a word such as Moodle
(Modular Object-Oriented Dynamic Learning Environment) at least when giving the
introduction.
4. Give appropriate explanation for all the tables and figures separately like figure 6, 7 and
table 7, 8, 9.
5. Authors are instructed to comprehensively proofread manuscript for typos and grammar
mistakes.

Experimental design

Improve the diagram of intelligent model which is figure 5 write models/tools of mining
instead of generally writing data mining.
Explain logic available in the table 2 last column.
Captions of figures are oversimplified.

Validity of the findings

conclusion are not well stated.
Among different models used for the work provide clarification about which model or
technique of mining gives better results in the context of proposed work

Reviewer 3 ·

Basic reporting

The idea of work is good because students' performance must be tracked in order to closely monitor all unfavorable barriers that may affect their academic progress.

Experimental design

1. A dataset containing 20 questions and 5 possible values has been used for the experiments. Is that dataset published/ approved or not?? If not then perform dataset validation by applying the previous algorithms or techniques as well. Additionally, if the dataset is matured and published then its source is missing in the script. It’s counsel to add the dataset source in script in order to corroborate the research foundation.
2. There are numerous algorithms designed for mining the data or possible outcomes.
What’s the reason behind choosing these particular algorithms such as Random Forest and Random Tree for gaining the desired results? Give the logic behind selecting these mining algorithms. Also, describe the reason for excluding the other techniques.

Validity of the findings

3. Comparative analysis this research study isn't available or performed. It's recommended to add the result of previous research studies in comparison. Rather than just comparing the used algorithms.

4. Detail of comparative analysis is required. Prove the improved results numerically and logically through a comparison table with existing research work which is done in prior years.

Additional comments

5. The Summary table of Literature should be added to the end of the literature review section that gives a quick tabular overview for a better understanding of the building blocks of the written Problem statement.

6. Additionally, add more papers in Literature from the latest years because only 1 paper is considered from the last two years.

---

## Round 0.2 · Minor Revisions

Please make "minor revisions" as per the given comments from reviewers.

Reviewer 1 ·

Basic reporting

My comments below only reflect my opinions about the changes. My prior positive comments remain as is.

1. The abstract is definitely in better shape, although it's still pretty long. I recommend the authors to reduce the size of the abstract to 4-5 sentences only. You can always include the details in the Introduction section.
2. The research objective has been clarified in Section 1, which I thank the authors for.
3. Lots of superfluous text has been removed, which I thank the authors for.
4. I thank the authors for providing the comparison between algorithms as figures instead of tables.

Experimental design

My comments below only reflect my opinions about the changes. My prior positive comments remain as is.

1. As requested, the authors have included analysis using other algorithms as well, and the analysis now appears complete to me.
2. I appreciate the authors including some text about the F-measure. While I would have wanted some more insight into its usage, the present form is acceptable to me.

Validity of the findings

I don't have any further comments. My concerns regarding adding more info on feature selection have been addressed to my satisfaction. I thank the authors for that.

Additional comments

While the text can be improved in several ways, I do not have any objection for publication in its current form.

Reviewer 2 ·

Basic reporting

Some comments are done. Still some changes are required like explanation of figure 7 and 8 are not good. and add more detail about 3, 4 and 5 figures.
At the end of literature add some discussion but summary is not required separate heading tabular form is good. Add more recent references.
carefully use capital letter words use the same pattern throughout paper. e.g Moodle

Experimental design

Methods describe with sufficient detail.

Validity of the findings

conclusion is not well stated . Write in more concise and impressive way and add future approach.

Reviewer 3 ·

Basic reporting

no comment

Experimental design

no comment

Validity of the findings

no comment

Additional comments

The authors have addressed all the concerns raised by the reviewers in the revised manuscript. Therefore, I recommend the revised manuscript for publication in the Journal.

---

## Round 0.3 · accepted · Accept

Based on the recommendations from the reviewers, the paper is accepted in its current form.

Reviewer 2 ·

Basic reporting

Author did job job in improving the paper.

Experimental design

investigations level is not very rigorous but the methodology is novel.

Validity of the findings

conclusions are well stated.